# Independent Impact of Gynoid Fat Distribution and Free Testosterone on Circulating Levels of N-Terminal Pro-Brain Natriuretic Peptide (NT-proBNP) in Humans

**DOI:** 10.3390/jcm9010074

**Published:** 2019-12-27

**Authors:** Małgorzata Chlabicz, Jacek Jamiołkowski, Marlena Paniczko, Paweł Sowa, Magda Łapińska, Małgorzata Szpakowicz, Natalia Jurczuk, Marcin Kondraciuk, Andrzej Raczkowski, Emilia Sawicka, Karol Adam Kamiński

**Affiliations:** 1Department of Population Medicine and Civilization Diseases Prevention, Medical University of Bialystok, 15-269 Bialystok, Poland; mchlabicz@op.pl (M.C.); jacek909@wp.pl (J.J.); m.paniczko@gmail.com (M.P.); sowa@umb.edu.pl (P.S.); magda.lapinska@umb.edu.pl (M.Ł.); malgorzata.szpakowicz@umb.edu.pl (M.S.); n.jurczuk@gmail.com (N.J.); marcin.kondraciuk@umb.edu.pl (M.K.); andrzej.raczkowski@umb.edu.pl (A.R.); emiliasawickak@gmail.com (E.S.); 2Department of Invasive Cardiology, Teaching University Hospital of Bialystok, 15-276 Bialystok, Poland; 3Department of Cardiology, Teaching University Hospital of Bialystok, 15-276 Bialystok, Poland

**Keywords:** natriuretic peptide, gynoid fat tissue, sex, population studies

## Abstract

Background: Natriuretic peptides (NPs), including brain natriuretic peptide (BNP), are neurohormones involved in the regulation of water-sodium balance and the maintenance of cardiovascular homeostasis. A higher concentration of NPs is observed in females, but the mechanism behind this difference has not been fully elucidated. Methods: Randomly chosen 255 volunteers from the general population were examined. Overall, 196 people without severe cardiovascular disease were included (mean age 48 years, 35.7% male). A comprehensive assessment was performed, including anthropometric measurements, N-terminal pro-brain natriuretic peptide (NT-proBNP), total testosterone (TT) and sex hormone-binding globulin (SHBG) concentration, transthoracic echocardiography (ECHO), and body composition analysis by direct dual-energy X-ray absorptiometry (DEXA). The univariate analysis adjusted by the known affecting factors determined which measurements were independently associated with NT-proBNP concentration. Results: NT-proBNP concentration was positively associated with gynoid fat mass, gynoid/total fat (G/TF) mass index, SHBG and negatively with android/gynoid (A/G) fat mass index, TT and calculated free testosterone (CFT) concentrations. Furthermore, body composition parameters remained independently associated with NT-proBNP levels even after adjusting for CFT and SHBG. Conclusion: In the population without severe cardiovascular disease, the NT-proBNP concentration is independently associated with lower availability of testosterone and higher gynoid fat distribution, which may explain higher NPs levels in females.

## 1. Introduction

Natriuretic peptides (NPs), including brain natriuretic peptide (BNP), are neurohormones involved in the regulation of water-sodium balance and maintenance of cardiovascular homeostasis. They exert cardioprotective effects by promoting vasodilatation, natriuresis and inhibit fibrosis counteracting the effects of the renin-angiotensin-aldosterone system. They are largely secreted by the myocardium in response to the increased wall stress, ischemia and hypertrophy [1,2]. N-terminal pro-brain natriuretic peptide (NT-proBNP) is the physiologically inactive 1–76 amino acid fragment, which is secreted with mature 32-amino acid BNP after cleavage from the prohormone [3,4,5]. The measurement of pathologically elevated plasma level of BNP and its co-released peptide NT-proBNP with a longer plasma half-life allows diagnosing cardiovascular diseases, especially heart failure (HF) [2,6]. A higher concentration of NPs is observed in older people, females, and in individuals with lower glomerular filtration rate (GFR) or atrial fibrillation (AF) while lower in obese people [7,8,9,10,11]. The mechanism behind these differences has not been fully elucidated. A prior study suggested that BNP levels were inversely correlated with lean body mass [9], or NT-proBNP concentration was inversely associated with visceral adipose tissue in both sexes and with subcutaneous adipose tissue only in women [12]. It has not been established whether there is a link between NPs concentration and regional fat distribution. Therefore, we have analyzed this relationship further extending the analysis to the effect of the testosterone bioavailability and sex hormone-binding globulin (SHBG), because of the established influence of gonadal steroids on body composition [13]. We hypothesized that the relation between plasma NT-proBNP concentration and differences in regional fat distribution could give an insight into the regulation of expression of NPs levels in the population sample without severe cardiovascular disease.

## 2. Materials and Methods

### 2.1. Study Population

The study was conducted in 2017–2018 in a representative sample of area residents aged between 20 to 77. According to the 2017 Central Statistical Office data, the number of residents in Bialystok was 297,300. Randomly selected residents (600) from the mayor’s office database were invited to participate in the study. Three people from the random group died before receiving an invitation, 255 individuals responded and were examined. Due to the history of ischaemic heart disease (IHD), previously diagnosed HF, cardiac surgery, valvular heart disease, AF or reduced left ventricular ejection fraction below 50% in the current echocardiography, 59 people were further excluded from the research group. As a result, 196 people (126 women and 70 men) were included in the study (Figure 1). 

### 2.2. Data Collection and Assays

The subjects’ medical records were obtained by the questionnaires filled at the time of entry. All study participants underwent a laboratory assessment and physical examination. Peripheral intravenous fasting blood samples were collected in the morning on a visit day. Plasma was stored at −80 °C until analyzed. For analysis, samples were thawed at 37 °C, prepared, and analyzed immediately. The concentration of NT-proBNP, total testosterone (TT) and sex hormone-binding globulin (SHBG) were determined by the electrochemiluminescence method on the Cobas e411 from ROCHE. The free testosterone (FT) was calculated (CFT) by published methods described by Ly and Handelsman [14] with the use of the following formulas:Model 1 EFT-lo = −6.593 + 19.304 TT + 0.056 SHBG − 0.0959 TT × SHBG,(1)
Model 2 EFT-hi = 52.65 + 24.4 TT − 0.704 SHBG − 0.0782 TT × SHBG − 0.0584 TT^2^,(2)

Model 1 was used for low testosterone <5 nmol/L, model 2 for high testosterone ≥5 nmol/ L (with TT and SHBG in nmol/L). The negative values of CFT were changed to 0.

The comprehensive assessment was performed: the anthropometric measurements including height, weight, circumferences of waist, hips and the measurement of handgrip strength. Blood pressure (BP) was measured while the participants were seated after a minimum rest period of 5 min by applying the oscillometric method and the Omron Healthcare Co., Ltd. MG Comfort instrument. The electrocardiography (ECG) was performed using AMEDTEC ECGpro CardioPart 12 USB (AMEDTEC Medizintechnik Aue GmbH, Aue, Germany). The transthoracic echocardiography (ECHO) was performed using B-mode ultrasound Vivid 9 (GE Healthcare, Chicago, IL, USA). In ECHO, the dimensions of the heart, left atrial (LA) volume, left ventricular ejection fraction (LVEF) using the biplane method were measured. Next left ventricular mass index (LVMI) was calculated, with left ventricular hypertrophy (LVH) defined as LVMI ≥ 115 g/m^2^ for men and ≥95 g/m^2^ for women, and left atrial volume index (LAVI) was calculated, with enlarged volume defined as >34 mL/m^2^. Diastolic dysfunction of the left ventricle was assessed based on the latest recommendation [15]. Body mass index (BMI) was calculated as weight in kilograms divided by height in meters squared. Body composition measurements were taken by dual-energy X-ray absorptiometry (DEXA), GE Healthcare Lunar, with total body mass divided into 3 compartments: bone, fat mass, and lean mass. By using the region of interest (ROI) program, total fat (TF), gynoid (G) and android (A) fat were measured automatically. The ranges for android ROI were: lower boundary at pelvis cut, upper boundary above pelvis cut by 20% of the distance between pelvis and neck cuts, lateral boundaries are the arm cuts. The ranges for gynoid ROI were: upper boundary below the pelvis cut line by 1.5 times the height of the android ROI, gynoid ROI height equal to 2 times the height of the android ROI, lateral boundaries were the outer leg cuts. The A/G ratio was calculated between the fat of the android (central) and fat of the gynoid (hip and thigh) regions. The G/TF ratio was calculated as the ratio between the gynoid fat and total fat. The analytical measurement range for NT-proBNP was 5–35.000 pg/mL, the concentrations below the detection threshold were taken as 50% of the minimum measurement range (one person).

### 2.3. Ethical Issues

Ethical approval for this study was provided by the Ethics Committee of the Medical University of Bialystok (Poland) on 31 March 2016 (approval number: R-I-002/108/2016). The study was conducted in accordance with the Declaration of Helsinki and all participants gave written informed consent.

### 2.4. Statistical Analysis

Descriptive statistics for quantitative variables were presented as means and standard deviations and as counts and frequencies for qualitative variables. Comparisons of continuous variables between subgroups were conducted using Mann-Whitney or Kruskall-Wallis tests. Associations between NT-proBNP and other clinical and biochemical variables were analyzed using simple and multiple linear regression models. Multiple regression models were adjusted for age, sex, GFR, BMI, history of hypertension, BP ≥ 140/90 mmHg (Model 1.), for age, sex, GFR, BMI, history of hypertension, BP ≥ 140/90 mmHg, CFT (Model 2.) or for age, sex, GFR, BMI, history of hypertension, BP ≥ 140/90 mmHg, CFT and SHBG (Model 3.). Regression models were presented using variable coefficients, *p*-values of Wald tests and coefficients of determination for the model (*R*^2^). Statistical hypotheses were verified at a 0.05 significance level. The IBM SPSS Statistics 20.0 statistical software (Armonk, NY, USA) was used for all calculations.

## 3. Results

The baseline characteristics of the study population are shown in Table 1.

The mean age was 47.8 ± 15 and 35.7% of participants were male. There was no difference in mean age between men and women. Overall, 23% of participants were obese (BMI ≥ 30 kg/m^2^) and 35.7% were overweight (BMI ≥ 25 kg/m^2^ and BMI < 30 kg/m^2^). The median NT-proBNP concentration was 59.81 pg/mL (Interquartile range (IQR) 27.72–93.88), with the cut-off values on the 90th percentile on the level 108.56 pg/mL for male and 156.87 pg/mL for female participants. Within the population, NT-proBNP levels were higher among women than men (*p* = 0.047) and they increased with age (*p* < 0.001, β = 2.084).

As shown in Table 2, there was no significant difference in mean total fat mass and android fat mass by DEXA between men and women. However, women had lower BMI, total lean mass, android lean mass, gynoid lean mass, visceral mass, while higher gynoid fat mass. Figure 2 shows NT-proBNP serum levels on a logarithmic scale stratified by CFT and gynoid fat. The NT-proBNP concentration was significantly higher in people with the lower CFT concentration (stratified by median CFT 8.68 pmol/L, *p* < 0.001) and with higher gynoid fat mass (stratified by median gynoid fat 4.04 kg, *p* = 0.017).

Table 3 shows the univariate predictors of higher NT-proBNP in general population without severe cardiovascular disease. Individuals with higher NT-proBNP concentration were older, more likely to be female, had lower GFR, had higher systolic blood pressure (BPs), declared history of hypertension or blood pressure medicine intake, had LVH and higher value of LAVI. A significant correlation between BMI and NT-proBNP concentration was not observed, a correlation coefficient was 0.197 in all groups, 0.359 and 0.145 for women and men, respectively. The other factors which positively correlated with NT-proBNP were parameters of carbohydrate metabolism: fasting glucose, 2-h glucose in oral glucose tolerance test (OGTT), hemoglobin A1c (HbA1c). Moreover, significantly associated with a higher concentration of NT-proBNP were: a higher SHBG concentration, a lower concentration of TT, CFT and dehydroepiandrosterone (DHEA). In the body composition analysis, a statistically significant correlation was demonstrated between a higher concentration of NT-proBNP and lower total lean mass, lower gynoid lean mass, higher gynoid fat mass and lower handgrip strength.

The association of NT-proBNP concentration with the above-mentioned parameters changed after adjustments for age, sex, GFR, BMI, history of hypertension, BP ≥ 140/90 mmHg (Model 1) were made. These factors are known for affecting the NT-proBNP level (Table 3). The parameters that remained independently associated with higher NT-proBNP were: lower testosterone (TT and CFT), higher SHGB level, higher gynoid fat mass, but also new parameters appeared: lower A/G fat mass index and higher G/TF mass index. Since there is an established influence of gonadal steroids on body composition in Model 2, we further extended the analysis by the effect of the testosterone. The higher concentration of SHBG, higher gynoid fat mass, higher G/TF mass index and lower A/G fat mass index were still significantly associated. In addition, a new significantly associated with a higher NT-proBNP concentration parameter appeared: lower visceral mass. In Model 3, these connections were slightly attenuated but remained statistically significant after adjusting for SHGB.

In stepwise backward linear regression analysis (Table 4), the strongest factors affecting NT-proBNP concentration were: age, sex, G/TF mass index, CFT and SHGB.

The gynoid (pear-shaped) body type remained positively associated with NT-proBNP concentration even after adjusting for gonadal steroids.

## 4. Discussion

The present study reports on the plasma NT-proBNP concentration in subjects without severe cardiovascular disease providing evidence that the bioavailability of the testosterone and body composition are independently associated with the NT-proBNP concentration, which may explain higher NPs levels in females. This is the first study that has shown an independent relationship between NT-proBNP concentration and gynoid fat distribution as well as testosterone concentration.

### 4.1. NT-proBNP Serum Levels in Relation to Gender

NPs are important in maintaining normal cardiac and metabolic status. Whether plasma concentrations of NPs within the normal range reflect cardio-metabolic health is unknown. This study confirmed that NT-proBNP levels are higher in females and increase with age in the general population [7,16,17,18,19]. The cut-off values for NT-proBNP on the 97.5th percentile reported by Galasko [17] were: 100 pg/mL and 172 pg/mL for men aged 45–59 and 60+, respectively, and for women 164 pg/mL and 225 pg/mL aged 45–59 and 60+, respectively. In this study, we did not divide the population by age because of the small cohort size. In Dallas Heart Study [16] the cut-off values for NT-proBNP on the 90th percentile were on the level 52.3 pg/mL for male and 103 pg/mL for female, but the maximum age of the subjects was 65 years, the mean age was 44 ± 9 years and the average BMI was 29.4 ± 6.7 kg/m^2^. In our study, the maximum age was 77 years, the mean age 47.83 ± 14.97 and the average BMI 26.28 ± 4.80 kg/m^2^, which could be associated with a higher concentration of NT-proBNP. According to Das [9], NT-proBNP levels are higher in females and increase with age in general population with cut-off values on the 90th percentile on the level 82.4 ng/L for male and 127.2 ng/L for female. The higher concentration of NT-proBNP in this research could be explained by different characteristics of the groups studied, as the participants were older, there were more women and less overweight and obese people. Correspondingly, the cohort size in the current study was smaller.

### 4.2. Impact of Clinical Factors on NT-proBNP

Our overall findings are in line with earlier studies. Daniels [20] showed, that individuals with higher NT-proBNP concentration were older, had lower BMI and GFR, were more likely to be female, had higher BPs, and displayed higher use of beta-blockers. Costello-Boerrigter [19] in potentially healthy patients (no history of CVD (cerebrovascular disease), pulmonary disease, renal disease, DM (diabetes mellitus), no systolic and diastolic dysfunction, no cardiovascular medication, in normal sinus rhythm) in univariate analysis showed, that the highest correlation coefficients with NT-proBNP concentration were increasing age, GFR, LAVI, LV dimension index and gender. The correlations were significant but weak for systolic BPs and BPd, BMI, and heart rate (HR). In the multivariable analysis in this group, female gender and older age were the strongest independent predictors of higher NT-proBNP. However, when the multivariable analysis was performed including the patients with reduced EF ≤40% and the general population: creatinine, GFR, BMI, and LV mass were also found to affect NT-proBNP independently. Galasko [17] reported that female gender and increasing age were the only independent predictors of increased NT-proBNP levels in subjects with no history of IHD, HF, DM, hypertension, CVD, peripheral vascular disease (PVD), loop diuretic usage, a history of heavy alcohol intake, LVEF < 50% and GFR < 60 mL/min/1.73 m^2^. In our study, in the population without severe cardiovascular disease, we also do not show a significant dependence between BMI and NT-proBNP concentration, but confirm a positive correlation with increasing age, female gender, lower GFR, higher BPs and history of hypertension and blood pressure medicine.

### 4.3. Impact of Obesity and Fat Tissue Distribution on NT-proBNP Concentration

The associations between NPs concentration and BMI or fat tissue distribution have been examined in epidemiologic studies. An inverse relationship has been demonstrated between NPs levels and BMI [21,22]. Several hypotheses have been proposed to explain this relationship. Wang [21] showed that higher BMI was associated with lower BNP levels, and they suggested that this inverse relationship could be due to lower expression of the natriuretic peptide clearance receptor (NPR-C) by adipose tissue resulting in increased clearance of BNP in obese individuals [21,23,24]. Supporting these theories, elevated NPR-C gene expression has been documented in the adipose tissue of individuals with obesity [25]. This, however, does not explain the lower concentration of NT-proBNP.

Adipocytes also express atrial natriuretic peptide receptor (NPR-A), which induces lipolysis. Thus, low NPs concentration may lead to depressed lipolysis further perpetuating obesity [24,26]. The physiologic studies have verified the beneficial metabolic effects of NPs. Infusion of atrial natriuretic peptide (ANP) at physiologic levels induced lipid mobilization from subcutaneous adipose tissue with increased lipide oxidation [27,28,29]. NPs can also regulate adipocyte proliferation and visceral adipose tissue expansion. A cell-culture model to study visceral adipose tissue growth revealed stimulation by angiotensin II and inhibition by ANP at physiological concentrations [30].

Another report from the Dallas Heart Study [9] suggested that nonclearance mechanisms should be more important because NT-proBNP cannot bind the NPR-C receptor, but similarly, it is inversely associated with BMI. It seems that impaired synthesis and secretion of NPs from the myocardium contribute to low NPs concentration in higher BMI [21]. Analysis of other forms of natriuretic peptides could supplement our data and provide more information on these mechanisms. Dallas Heart Study [9] showed that lean body mass was more strongly associated with NT-proBNP than fat mass and suggested that this effect could be mediated by sex steroid hormones that parallelly influence NPs synthesis as well as body composition. Sarzani [24] showed that an abdominal type of obesity predicted lower NPs levels in women, even after adjustment for BMI, which is consistent with our study.

### 4.4. Impact of Gender and Steroid Hormones on NT-proBNP

Many studies independently reported other factors that contribute to a higher concentration of NPs in the female population [7,8,31], but the mechanism responsible for this phenomenon is still not fully elucidated. Women, compared to men, have a higher body fat percentage [32]. For the same BMI, a female has about 10% higher body fat compared to a man [33], which may suggest a lower NT-proBNP concentration. In fact, it is quite opposite. Subcutaneous white adipose tissue (SAT) store about 80–90% of total body fat, mostly in the abdominal, subscapular, gluteal and femoral areas. Intra-abdominal fat depots contain visceral adipose tissue (VAT), which is associated with digestive organs [34,35,36,37,38].

There is an established influence of gonadal steroids on body composition, that could play a role in a potential modification. Sex differences arise during puberty. The weight gain in boys is due to increases in lean mass while in girls due to increases in fat mass [39,40,41]. Menopause occurs in the redistribution of adipose tissue towards a more android type, presumably due to the fall in estrogen levels and rise in testosterone [42,43,44]. As testosterone concentration declines with age, visceral adiposity also increases in men [13,45,46]. Bann [13] suggested a role of steroid hormones in regulating fat distribution showing that testosterone level was negatively associated with fat mass in men and positively associated with women. Moreover, the decline in testosterone was not associated with lean mass in either sex, and SHBG was inversely associated with fat mass and A/G ratio in both men and women. Observed BNP changes after puberty also suggested that steroid hormones mediate gender differences in BNP [47].

Redfield et al. [8] identified a relationship between hormone replacement therapy (HRT) and concentration of BNP and suggested that BNP production may be sensitive to estrogen regulation. Another study [48] demonstrated an increase in mean BNP concentration after three months of transdermal estradiol administration. In our study, we do not analyze estradiol concentration nor use of HRT. Chang [49] showed that testosterone was independently and inversely associated with BNP and NT-proBNP, and suggested that testosterone, not estradiol, mediates gender differences in NPs. Lam [50] showed that lower testosterone level and higher SHBG levels correlated with increasing NT-proBNP concentration and receiving hormonal contraceptives (HC) contribute to a higher level of NT-proBNP in women. In our study, we did not analyze the relationship between using HRT or hormonal contraception and concentration of NT-proBNP, because of a small number of women using these therapies.

### 4.5. Possible Mechanism

Gluteal-femoral adipose tissue may play an active role in metabolism [38]. In male mice, transplantation of inguinal SAT as compared to epididymal (visceral depot) inside the abdominal cavity causes less body weight gain and better glucose tolerance [51,52]. Other studies suggested that subcutaneous women adipocytes had higher lipoprotein lipase (LPL) activity [53], lipid synthesis [54] and insulin-stimulated glucose uptake [55] compared to men. It may also affect the metabolism of NPs, which could explain the phenomenon that gluteal-femoral adipose tissue lessens metabolic risk factors.

### 4.6. Limitation

A relatively small number of enrolled patients prevented an analysis with a breakdown into a population with normal body weight, overweight, and obesity. Other forms of natriuretic peptides were not measured. Measurements of estradiol and estrogen were not available in this study.

## 5. Conclusions

NPs concentration is independently associated with body composition, especially with gynoid fat tissue and bioavailability of testosterone, which may explain the higher concentration of NT-proBNP in the female population. The present study does not provide any explanation for this relationship. Further studies will be needed to confirm these findings and understand the associated mechanism.

## Figures and Tables

**Figure 1 jcm-09-00074-f001:**
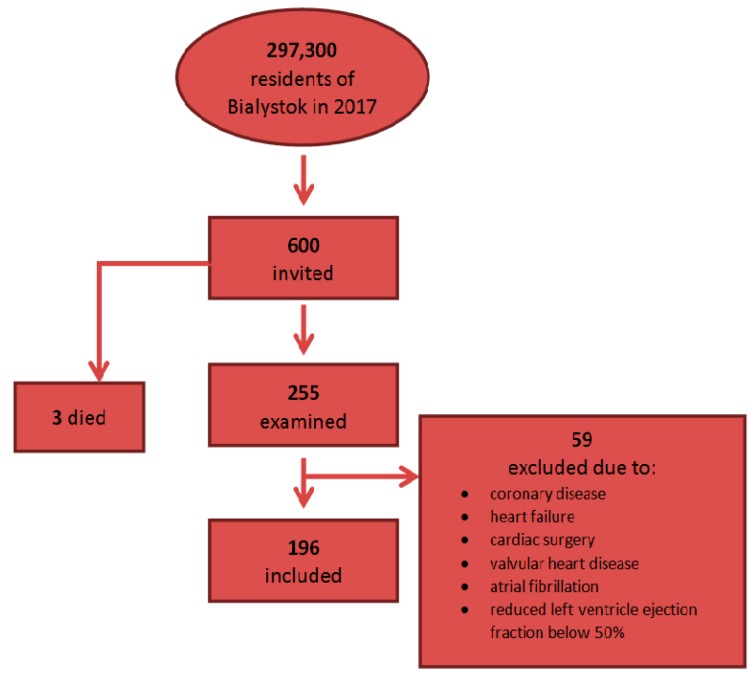
Analyzed cohort of 196 included individuals.

**Figure 2 jcm-09-00074-f002:**
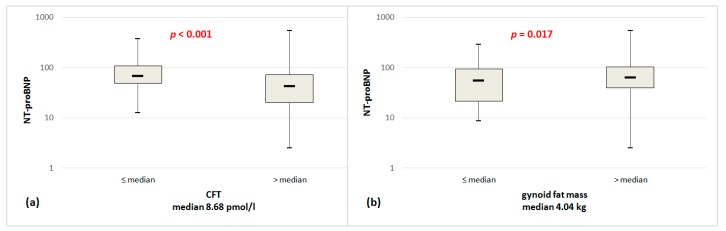
NT-proBNP serum levels in a population stratified by CFT (**a**), gynoid fat mass (**b**) on a logarithmic scale; comparisons between subgroups were conducted using Mann-Whitney tests; NT-proBNP, N-terminal pro-brain natriuretic peptide; CFT, calculated free testosterone.

**Table 1 jcm-09-00074-t001:** Characteristics of the cohort.

Variable	Value (*n* = 196)
Age, years	47.83 ± 14.97
Male sex	70 (35.7)
BMI, kg/m^2^	26.28 ± 4.80
BMI 25–29.99 kg/m^2^	70 (35.7)
BMI ≥ 30 kg/m^2^	45 (23.0)
NT-proBNP, pg/mL	72.59 ± 67.15
hs-TnT, pg/mL	6.02 ± 3.36
Diastolic dysfunction of left ventricle	19 (9.7)
LVMI, g/m^2^	82.01 ± 22.38
LAVI, mL/m^2^	22.88 ± 7.09
Creatinine, μmol/L	75.90 ± 16.71
GFR, mL/min/1.73 m^2^	103.63 ± 31.42
BPs, mmHg	121.28 ± 16.72
BPd, mmHg	81.60 ± 10.49
BP ≥ 140 and/or ≥ 90 mmHg	42 (21.8)
HR, bpm	73.68 ± 11.66
History of hypertension	56 (28.9)
History of diabetes	13 (6.7)
Currently smoking	47 (24.0)

The data is shown as *n* (%), mean ± SD. SD, standard deviation; BMI, body mass index; kg, kilogram; m^2^, square meter; NT-proBNP, N-terminal pro-brain natriuretic peptide; hs-TnT, high-sensitivity cardiac troponin T; LVMI, left ventricular mass index; g, gram; LAVI, left atrial volume index; mL, milliliter; GFR, glomerular filtration rate Cockcroft-Gault Equation; BPs, systolic blood pressure; BPd, diastolic blood pressure; mmHg, millimeters of mercury; HR, heart rate; bpm, beats per minute.

**Table 2 jcm-09-00074-t002:** Gender group characteristics.

Variables	Woman	Man	*p*-Values
Number of subjects	126 (64.29)	70 (35.71)	
Age, years	48.25 ± 15.05	47.07 ± 14.89	0.493
BMI, kg/m^2^	25.82 ± 5.13	27.11 ± 4.04	0.035
NT-proBNP, pg/mL	79.69 ± 53.73	59.81 ± 79.00	<0.001
hs-TnT, pg/mL	5.40 ± 3.40	7.12 ± 3.02	<0.001
Creatinine, μmol/L	68.93 ± 13.08	88.35 ± 15.25	<0.001
GFR, mL/min/1.73 m^2^	99.77 ± 32.66	110.53 ± 27.96	0.003
HbA1c, %	5.44 ± 0.48	5.51 ± 0.50	0.398
TT, ng/mL	0.20 ± 0.13	4.53 ± 1.89	<0.001
CFT, pmol/L	5.52 ± 5.34	229.74 ± 91.12	<0.001
SHBG, nmol/L	69.25 ± 34.25	40.95 ± 19.60	<0.001
LVMI, g/m^2^	74.78 ± 18.88	95.86 ± 22.16	<0.001
LVMI, ≥95 g/m^2^ women, ≥115 g/m^2^ men	17 (15.0)	10 (16.9)	0.744
LAVI, mL/m^2^	21.98 ± 6.65	24.58 ± 7.63	0.031
LAVI, >34 mL/m^2^	6 (5.5)	7 (12.1)	0.127
Diastolic dysfunction of left ventricle	16 (12.7)	3 (4.3)	0.060
BPs, mmHg	116.73 ± 15.91	129.66 ± 14.91	<0.001
BPd, mmHg	80.04 ± 9.61	84.47 ± 11.47	0.010
BP ≥ 140 or/and ≥90, mmHg	20 (16.0)	22 (32.4)	0.009
HR, bpm	74.03 ± 10.58	73.04 ± 13.49	0.465
Total fat mass, kg	26.23 ± 9.38	23.72 ± 8.51	0.136
Total lean mass, kg	41.51 ± 5.03	57.54 ± 7.15	<0.001
Android fat mass, kg	2.11 ± 1.15	2.45 ± 1.23	0.055
Android lean mass, kg	2.86 ± 0.37	3.83 ± 0.54	<0.001
Gynoid fat mass, kg	4.47 ± 1.4	3.26 ± 1.12	<0.001
Gynoid lean mass, kg	6.22 ± 0.75	8.53 ± 1.28	<0.001
Visceral mass, kg	0.81 ± 0.68	1.59 ± 1.05	<0.001
History of hypertension	34 (27.2)	22 (31.9)	0.491
History of blood pressure medicine	33 (26.2)	23 (32.9)	0.322
History of diabetes	7 (5.6)	6 (8.7)	0.418
Current smoking	28 (22.22)	19 (27.14)	0.897

The data is shown as *n* (%), mean ± SD. SD, standard deviation; BMI, body mass index; kg, kilogram; m, meter; NT-proBNP, N-terminal pro-brain natriuretic peptide; hs-TnT, high-sensitivity cardiac troponin T; GFR, glomerular filtration rate Cockcroft-Gault Equation; mL, milliliter; HbA1c, hemoglobin A1c; TT, total testosterone; CFT, calculated free testosterone; SHBG, sex hormone-binding globulin; LVMI, left ventricular mass index; g, gram; LAVI, left atrial volume index; BPs, systolic blood pressure; BPd, diastolic blood pressure; mmHg, millimeters of mercury; HR, heart rate; bpm, beats per minute.

**Table 3 jcm-09-00074-t003:** Results of NT-proBNP analysis in the general population.

Variables	Univariate Analysis			Model 1			Model 2			Model 3		
	B	*p*	*R* ^2^	B	*p*	Adjuste *R*^2^	B	*p*	adjusted *R*^2^	B	*p*	Adjusted *R*^2^
Age, years	2.084	<0.001	0.216	-	-	-	-	-	-	-	-	-
Gender, male	−19.885	0.047	0.020	-	-	-	-	-	-	-	-	-
GFR, mL/min/1.73 m^2^	−0.713	<0.001	0.112	-	-	-	-	-	-	-	-	-
BMI, kg/m^2^	1.294	0.197	0.009	-	-	-	-	-	-	-	-	-
BPs, mmHg	0.818	0.005	0.041	-	-	-	-	-	-	-	-	-
BPd, mmHg	0.020	0.965	0.000	-	-	-	-	-	-	-	-	-
BP ≥ 140 and/or ≥90 mmHg	31.997	0.006	0.038	-	-	-	-	-	-	-	-	-
WHR	1.780	0.973	0.000	−81.088	0.372	0.232	−109.141	0.222	0.265	−88.341	0.316	0.292
HR, bpm	0.055	0.896	0.000	0.299	0.427	0.232	0.307	0.405	0.262	0.278	0.442	0.290
LVMI, g/m^2^	0.360	0.126	0.014	0.101	0.727	0.209	0.023	0.937	0.233	−0.008	0.976	0.257
LVMI, ≥95 g/m^2^ women, ≥115 g/m^2^ men	29.355	0.042	0.024	8.276	0.562	0.210	5.984	0.671	0.234	4.044	0.771	0.258
LAVI, mL/m^2^	1.576	0.036	0.026	1.122	0.111	0.250	1.043	0.1135	0.267	1.132	0.097	0.300
LAVI, >34 mL/m^2^	3.227	0.872	0.000	−0.948	0.958	0.238	−4.344	0.807	0.257	−0.529	0.976	0.287
Diastolic dysfunction of left ventricle	20.492	0.207	0.008	−3.407	0.824	0.226	0.884	0.953	0.256	−1.350	0.928	0.285
Fasting glucose, mg/dL	1.052	0.002	0.050	0.302	0.433	0.232	0.358	0.343	0.263	0.434	0.242	0.293
2-h glucose, mg/dL	0.235	0.021	0.031	−0.024	0.815	0.252	−0.011	0.912	0.274	−0.027	0.781	0.334
HOMA-IR	1.559	0.628	0.001	−2.750	0.446	0.231	−2.908	0.411	0.262	−2.878	0.406	0.291
HbA1c, %	29.227	0.004	0.044	−6.112	0.596	0.233	−5.577	0.621	0.262	−1.453	0.896	0.291
DHEA-s, µg/dL	−0.184	<0.001	0.111	−0.003	0.946	0.229	−0.020	0.693	0.260	0.001	0.980	0.288
TT, ng/mL	−5.855	0.003	0.047	−9.371	0.015	0.234	-	-	-	-	-	-
CFT, pmol/L	−0.145	<0.001	0.066	−0.221	0.004	0.263	-	-	-	-	-	-
SHBG, nmol/L	0.433	0.003	0.045	0.394	0.010	0.257	0.430	0.004	0.292	-	-	-
Total fat mass, kg	0.001	0.096	0.014	0.001	0.455	0.229	0.000	0.777	0.257	9.27	0.948	0.286
Total lean mass, kg	−0.001	0.004	0.042	0.000	0.846	0.227	0.000	0.714	0.257	0.001	0.578	0.287
Android fat mass, kg	0.005	0.194	0.009	−0.002	0.861	0.227	−0.009	0.351	0.260	−0.008	0.377	0.289
Android lean mass, kg	−0.011	0.133	0.012	0.008	0.557	0.228	0.003	0.803	0.257	0.007	0.594	0.287
Gynoid fat mass, kg	0.007	0.041	0.021	0.019	0.004	0.261	0.017	0.009	0.285	0.015	0.028	0.305
Gynoid lean mass, kg	−0.009	0.004	0.043	0.000	0.969	0.226	0.001	0.884	0.257	0.002	0.728	0.286
Visceral mass, kg	0.005	0.311	0.005	−0.017	0.073	0.240	−0.024	0.014	0.282	−0.023	0.015	0.309
A/G fat mass ratio	1.956	0.928	0.000	−90.002	0.014	0.252	−108.543	0.003	0.294	−93.035	0.010	0.312
G/TF mass ratio	129.221	0.451	−0.002	874.08	0.001	0.275	874.040	<0.001	0.306	772.721	0.002	0.323
Handgrip strength test, max	−1.487	<0.001	0.100	−0.801	0.070	0.243	−0.778	0.073	0.272	−0.742	0.081	0.300
History of hypertension	43.278	<0.001	0.085	-	-	-	-	-	-	-	-	-
History of hypertension treatment	41.406	<0.001	0.078	-	-	-	-	-	-	-	-	-

NT-proBNP, N-terminal pro-brain natriuretic peptide; GFR, glomerular filtration rate Cockcroft-Gault Equation; mL, milliliter; BMI, body mass index; kg, kilogram; m, meter; BPs, systolic blood pressure; BPd, diastolic blood pressure; WHR, waist-hip ratio; HR, heart rate; mmHg, millimeters of mercury; bpm, beats per minute; LVMI, left ventricular mass index; LAVI, left atrial volume index; HOMA-IR, homeostatic model assessment of insulin resistance; HbA1c, hemoglobin A1c; DHEA, dehydroepiandrosterone; SHBG, sex hormone-binding globulin; TT, total testosterone; CFT, calculated free testosterone; A, android; G, gynoid; TF, total fat; Model 1: adjusted for age, sex, GFR, BMI, history of hypertension, BP ≥ 140/90 mmHg; Model 2: model 1 + additional adjustment for CFT; Model 3: model 2 + additional adjustment for SHBG.

**Table 4 jcm-09-00074-t004:** Results of stepwise backward linear regression analysis of relative NT-proBNP concentration with baseline variables.

Variables	Full Model			Final Model		
	B	*p*	*R* ^2^	B	*p*	*R* ^2^
Age, year	1.926	<0.001	0.323	2.447	<0.001	0.326
Gender, male	71.641	0.001	0.323	72.909	<0.001	0.326
G/TF mass ratio	772.721	0.002	0.323	659.928	0.004	0.326
CFT, pmol/L	−0.235	0.002	0.323	−0.245	0.001	0.326
SHBG, nmol/L	0.349	0.020	0.323	0.359	0.012	0.326
BMI, kg/m^2^	1.035	0.476	0.323	-	-	-
GFR, mL/min/1.73 m^2^	−0.145	0.488	0.323	-	-	-
History of hypertension	12.034	0.302	0.323	-	-	-
BP ≥ 140 and/or ≥90 mmHg	12.425	0.247	0.323	-	-	-

Variables entered into the model: age, sex, gynoid/total fat (G/TF) mass ratio, calculated free testosterone (CFT), sex hormone-binding globulin (SHBG), body mass index (BMI), glomerular filtration rate Cockcroft-Gault Equation (GFR), history of hypertension, blood pressure (BP) ≥ 140 and/or ≥90 mmHg.

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
