# Peer review of "Independent Impact of Gynoid Fat Distribution and Free Testosterone on Circulating Levels of N-Terminal Pro-Brain Natriuretic Peptide (NT-proBNP) in Humans"

_jcm, 2019, doi:10.3390/jcm9010074_

Round 1

Reviewer 1 Report

The article is well-written and I have no further concerns.

Author Response

We would like to thank the reviewer for insightful comments.

Reviewer 2 Report

The revised manuscript has good readibility, so the authors have been successful in their re=working of the English.  I find the title of the paper to be overly complicated and suggest it reads:  Independent impact of gynoid fat distribution and testostererone on circulating levels of N-terminal pro-brain natriuretic peptide (NT-proBNP) in humans. (note 'levels' not 'level')

Author Response

Dear reviewer.

I would like to thank for kind reviews. We agree that the new version of the title proposed by the reviewer is better. Therefore I am pleased to resubmit an original research article entitled “Independent impact of gynoid fat distribution and free testostererone on circulating levels of N-terminal pro-brain natriuretic peptide (NT-proBNP) in humans.” 

This manuscript is a resubmission of an earlier submission. The following is a list of the peer review reports and author responses from that submission.

Round 1

Reviewer 1 Report

In the current manuscript, Chlabicz et al. analyzed and reported the NT-proBNP correlations with age, gender and fat distribution in a cohort without severe CV disease. Though showed some statistical significance, the study did not present good novelty and knowledge to the field. A few comments for the authors' consideration.

age, gender, obesity and fat deposition have been extensively reported to contribute to natriuretic peptide levels in the general community, and thus the current study did not add much value to our knowledge in the field. the possible novel concept is the testosterone and SHBG. However, the fact is Lam et al reported the correlations of these two hormones with natriuretic peptide in a large cohort in 2011 JACC. spelling error, line 29, by known not by know. Other forms of natriuretic peptides can be measured to supplement the NT-proBNP data presented here.

Reviewer 2 Report

The study described here is a well conducted collection of data from healthy adult humans.  These have been examined for correlations and they provide a useful contibution to the literature on the relationship of NT-proBNP levels in the circulation to other factors. I have no concerns with the merit of the paper but consider the title to be misleading.  It is not an investigation into 'causes'.  A more approptiate title could be something like - 'Sex differences in circulating N-terminal pro-brain natriuretic peptide (NT-proBNP) are associated with availabiulity of testosterone and higher gynoid fat distribution.'

Apart from numerous minor examples of incorrect English throughout the text, there are specific errors.  Fig 2 caption should have (b). Table 4 footer should have 'gynoid', not 'gynoig'.